# CREDIT-BASED SELF ORGANIZING MAPS: TRAINING DEEP TOPOGRAPHIC NETWORKS WITH MINIMAL PERFORMANCE DEGRADATION

**Amir Ozhan Dehghani**[1,3]   **Xinyu Qian**[2]   **Asa Farahani**[4]   **Pouya Bashivan**[1,2,3]

[1] Department of Physiology, McGill University
[2] Department of Computer Science, McGill University
[3] Mila, Université de Montréal
[4] Montreal Neurological Institute, McGill University

## ABSTRACT

In the primate neocortex, neurons with similar function are often found to be spatially close. Kohonen's self-organizing map (SOM) has been one of the most influential approaches for simulating brain-like topographical organization in artificial neural network models. However, integrating these maps into deep neural networks with multitude of layers has been challenging, with self-organized deep neural networks suffering from substantially diminished capacity to perform visual recognition. We identified a key factor leading to the performance degradation in self-organized topographical neural network models: the discord between predominantly bottom-up learning updates in the self-organizing maps, and those derived from top-down, credit-based learning approaches. To address this, we propose an alternative self organization algorithm, tailored to align with the top-down learning processes in deep neural networks. This model not only emulates critical aspects of cortical topography but also significantly narrows the performance gap between non-topographical and topographical models. This advancement underscores the substantial importance of top-down assigned credits in shaping topographical organization. Our findings are a step in reconciling topographical modeling with the functional efficacy of neural network models, paving the way for more brain-like neural architectures. The code is available at https://github.com/BashivanLab/CBSOM.git.

## 1 INTRODUCTION

The neocortex is a highly structured biological system, with many of its functional capabilities tied to distinct anatomical regions. Processes such as visual, auditory, and somatosensory perception, as well as motor planning, are localized to specific cortical areas (Hubel & Wiesel, 1962; Humphries et al., 2010; Wong et al., 1978; Foerster, 1936). This structured organization extends to more granular scales within each cortical area, where neurons are grouped based on specific preferences. For example, in the visual cortex, neurons are organized according to the visual features they process (Hubel & Wiesel, 1962), while in the motor cortex, they are arranged according to the body parts they control (Foerster, 1936).

In the past decade, the reemergence of deep neural networks as tools for modeling neural computations has sparked a new wave of popularity in computational neuroscience (Yamins et al., 2014; Khaligh-Razavi & Kriegeskorte, 2014). Deep neural networks such as convolutional networks were shown to capture functional aspects of the neural code across the hierarchy of visual cortex in human (Khaligh-Razavi & Kriegeskorte, 2014; Ratan Murty et al., 2021b) and non-human primates (Yamins et al., 2014; Bashivan et al., 2019) as well as rodents (Nayebi et al., 2021). However, despite this remarkable representational similarity, it is widely observed that the arrangement of units (filters in convolutional networks) in these models is arbitrary and therefore they lack any topographical organization, unlike those in many animal brains (Lee et al., 2020; Bashivan et al., 2022).

It is widely believed that the multi-scale cortical organization is the result of mapping high dimensional neural representations onto the two-dimensional cortex through self-organization of neurons (Kohonen, 1982; Doshi & Konkle, 2023). Topographical organization is primarily thought to minimize neural wiring across cortical circuits, optimizing physical size and boosting capacity (Jacobs & Jordan, 1992), though recent work suggests it may also aid in learning robust representations (Qian et al., 2024). Beyond their functional roles, topographical models are valuable for therapeutic applications, such as guiding epilepsy surgeries by predicting cortical organization to help avoid resecting critical patches, like those involved in face recognition Schrimpf et al. (2024). In AI, inducing local structure in networks could enhance interpretability and aid in mitigating harmful behaviors by targeting specific clusters linked to undesired outputs.

During the past several years, a number of studies have proposed different approaches towards topographical deep neural networks that not only mimic the brain's activity patterns but also its topographical organization (Blauch et al., 2022; Margalit et al., 2024; Finzi et al., 2023; Doshi & Konkle, 2023). These approaches make use of various topography-inducing learning objectives (Blauch et al., 2022; Margalit et al., 2024) and learning rules (Doshi & Konkle, 2023) to impose structure across the DNN units' weight parameters in order to encourage topographically organized unit selectivity in each layer. In turn, while these models qualitatively capture facets of cortical topography, *they invariably suffer from substantial blows to their task performances (i.e. object recognition)*. It is therefore natural to question whether topography can emerge without a significant decline in performance.

We hypothesized that the source of such blows to task performance in topographical neural network models is the conflict between top-down gradient based learning with local topography-inducing objective functions or learning rules used in prior approaches (Kohonen, 1982; Von der Malsburg, 1973). Hence, we reasoned that *aligning the topography according to top-down measures of credit could potentially reduce such conflict and lead to topographical neural network models with minimal reduction in task performance*. To this end, we propose a new variation of the well known Self Organizing Map (SOM) (Kohonen, 1982), where the competitive learning rule is guided by the credit assigned to each unit based on its contribution to minimizing an ecologically-relevant task loss (Figure 1a). We adapted this method for use in deep neural networks (particularly convolutional networks) and parameter training with gradient descent and analyzed the resulting network's representations and topographical organization.

Our main contributions are as follows:

- We introduce a new algorithm for learning topographic deep neural networks called Credit-Based Self Organizing Maps (CB-SOM). Using this algorithm, we train a deep convolutional network in which filters in all layers are topographically organized.

- Despite remaining performance gap between topographical and non-topographical models, CB-SOM significantly improves object recognition compared to previous topographical models like Kohonen's SOM, while achieving strong topographical alignment with macaque and human ventral visual cortices across primary and high-level regions.

- We demonstrate enhanced alignment between CB-SOM representations and the neural activity in the macaque and human brains.

## 2  RELATED WORK

**Topographical models based on Hebbian learning.** Early attempts to replicate cortex-like topographical organization employed shallow neural networks, typically consisting of just a single hidden layer, with symbolic inputs such as bars of varying orientations (Kohonen, 1982; Von der Malsburg, 1973; Linsker, 1986; Swindale, 1996). These models were inspired by the lateral connections between neurons on the cortical surface and their parameters were trained unsupervised using different variations of the Hebbian learning rule. More recent work extended self-organizing approaches to additionally consider task-relevant information such as object labels during learning (Hartono et al., 2015; Hartono, 2020). While many of these models could replicate the organization of the neurons in the primary visual cortex with high fidelity, their utility was largely confined to early sensory cortices. This limitation was likely due to the inefficiency of local learning rules in constructing the representational hierarchies found in modern deep networks trained with backpropagation. Despite these challenges, recent work (Doshi & Konkle, 2023) has shown that applying some of these local learning rules, such as Kohonen's self-organizing maps, to pre-learned fea-

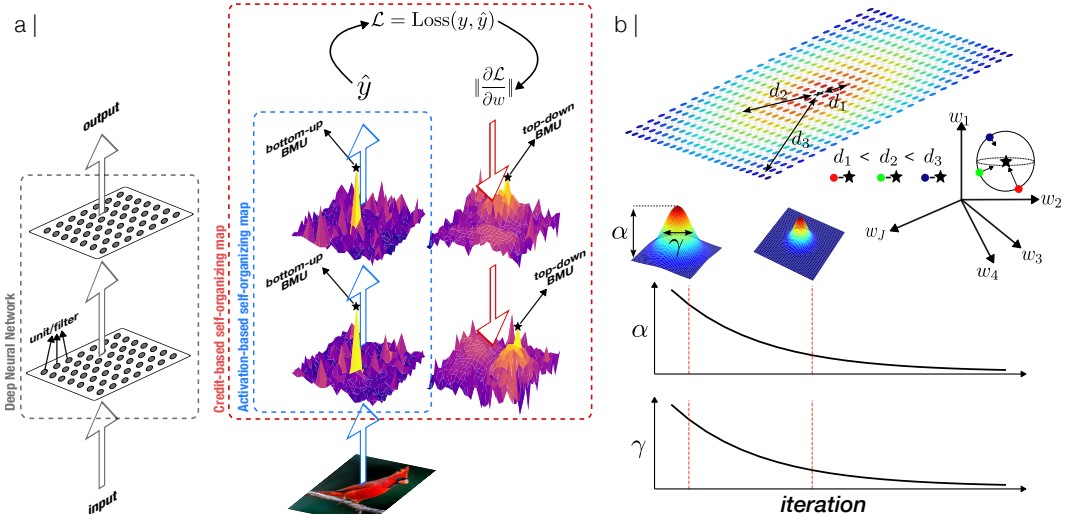

Figure 1: **A schematic of AB-SOM and CB-SOM models and their respective winner unit/filter selection algorithm.** Filters are arranged on a 2D map where in each iteration, one filter is chosen as the Best-Matching Unit (BMU). **a)** In AB-SOM, the filter that produces the highest response to a given input is chosen as the BMU (averaged across batch and spatial dimensions), whereas in the CB-SOM, the filter with the highest norm of gradient (i.e. $\frac{\partial \mathcal{L}}{\partial w}$) is chosen as BMU; **b)** (top) All parameters are updated according to a Gaussian neighborhood function. The neighborhood function weighs the parameter update in each filter as a function of the distance between the respective filter's distance to the BMU (star) on the 2D map. The Gaussian neighborhood function exponentially decays the effective weight updates for filters that are farther from the BMU. (bottom) The slope of the neighborhood function and the SOM learning rate are both follow an exponentially decaying schedule during training.

ture representations can yield topographical organizations that closely resemble those in the human brain. However, this approach necessitates a division of the learning process into two distinct and sequential phases — first learning non-topographical representations, followed by a topographical restructuring phase.

**Deep topographical neural networks.** In recent years, several approaches have been developed for training deep neural networks with topographical organization across multiple layers. Most of these methods rely on auxiliary learning objectives designed to promote a structured similarity between units/filters based on their physical distance within a predetermined layout (Lee et al., 2020; Blauch et al., 2022; Lu et al., 2023; Margalit et al., 2024). (Blauch et al., 2022) draws inspiration from the wiring cost optimization theory of cortical topography (Jacobs & Jordan, 1992), formulating a loss function that captures the cumulative connection strength between units across different layers as a function of their distance. By minimizing this loss during training, they reduce the strength of connections between distant units, resulting in clusters of model units that are selective for different object categories. Lee et al. (2020) and Margalit et al. (2024) introduce a loss function that encourages the pairwise correlation of activity between units to follow an exponentially decaying pattern, mirroring what is observed in the high-level visual cortex of primates. In contrast, Lu et al. (2023) adopts a comparable approach but enforces similarity in the weights of the units rather than their activations. Another recent study showed that incorporating lateral connections between filters in convolutional layers can replicate various aspects of cortical topography within these networks (Qian et al., 2024). Despite these advances, these models typically exhibit large drops in performance compared to non-topographical models (15-37% relative drop in object recognition accuracy).

## 3 METHODS

### 3.1 ACTIVATION-BASED SELF-ORGANIZING MAPS (AB-SOM)

AB-SOM is an adaptation of Kohonen's self-organizing maps for deep convolutional networks. It involves three steps:

1. *Cortical Sheet Arrangement*: Filters within each layer are positioned on a 2-dimensional *simulated cortical sheet* with predefined 2D coordinates $(i, j)$.

2. *Best Matching Unit (BMU) Selection*: Filter activities (i.e. feature maps) are averaged across a mini-batch of input images $\mathbf{x}$ and spatial dimensions. The filter $\mathbf{w}_{ij}$ with the highest average activation is selected as the BMU:

$$\text{BMU} = \arg\max_{ij} |\mathbf{w}_{ij}\mathbf{x}| \tag{1}$$

3. *Weight Updates*: Filter weights are updated based on their distance to the BMU on the cortical sheet, scaled by a learning rate $\alpha(t)$ and a neighborhood function $\gamma_{c,ij}(t)$, which is centered at the BMU position.

$$\mathbf{w}_{ij}(t+1) = \mathbf{w}_{ij}(t) + \alpha(t) \cdot \gamma_{c,ij}(t) \cdot (\mathbf{w}_c(t) - \mathbf{w}_{ij}(t))) \tag{2}$$

Here, $\mathbf{w}_c(t)$ is the BMU's weight, and $\mathbf{w}_{ij}(t)$ is the weight of the filter at position $(i, j)$. The neighborhood function $\gamma_{c,ij}(t)$ is defined as:

$$\gamma_{c,ij}(t) = \exp\left(-\frac{\|\mathbf{p}_c - \mathbf{p}_{ij}\|^2}{2\sigma(t)^2}\right) \tag{3}$$

where $\mathbf{p}_c$ and $\mathbf{p}_{ij}$ are the positions of the BMU and the filter at $(i, j)$, and $\sigma(t)$ is the variance. Both $\alpha(t)$ and $\sigma(t)$ decay over time. The neighborhood function was proposed as an approximate model of the functional interaction between nearby cortical neurons whose connectivity strengths decay exponentially as a function of cortical distance (Tuevo Kohonen, 1990) (see more details in section A.1).

## 3.2 CREDIT-BASED SELF-ORGANIZING MAPS (CB-SOM)

This is a variation of self-organizing maps where the competitive learning process is guided by the utility or credit assigned to different units/filters as opposed to activity level which is used in AB-SOM. The weight update procedure consists of the following steps:

1. *Cortical Sheet Arrangement*: Similar to AB-SOM,filters within each layer are positioned on a 2-dimensional *simulated cortical sheet*.

2. *Best Matching Unit (BMU) Selection*: The input is passed through all model layers in order to compute the model output. The model output in conjunction with the object label corresponding to the input pattern are used to compute the object recognition loss $\mathcal{L}$. The gradient of loss function with respect to each weight parameter is calculated to estimate each weight's importance on reducing the loss. Within each layer, the filter with the highest magnitude of gradient ($L_2$ norm of the cross-entropy loss gradient for each filter computed per batch of images) is selected as the winning unit/filter.

$$\text{BMU} = \arg\max_{ij} \left\|\frac{\partial \mathcal{L}}{\partial w_{ij}}\right\|^2 \tag{4}$$

3. *Weight Updates:* All weights are first updated with gradient descent to reduce the task loss. After this phase, the errors between each layer's winning filter's weight and other filters' weights within that layer are calculated and all weights are updated according to Eq. 2, similar to AB-SOM.

## 3.3 MODEL TRAINING

Each of the two self-organizing rules were incorporated into the ResNet-18 convolutional neural network architecture (He et al., 2016). Self-organization in each model was performed according to the corresponding rule (AB- or CB-SOM) at every layer of the network and every training step. The self-organization procedure was performed in tandem with the top-down gradient-based learning using stochastic gradient descent and an object recognition loss (cross-entropy loss) on the ILSVRC-2012 (ImageNet Large-Scale Visual Recognition Challenge) dataset (Deng et al., 2009) for 90 epochs with a batch size of 256. The initial learning rate was set to 0.1 and were decrease

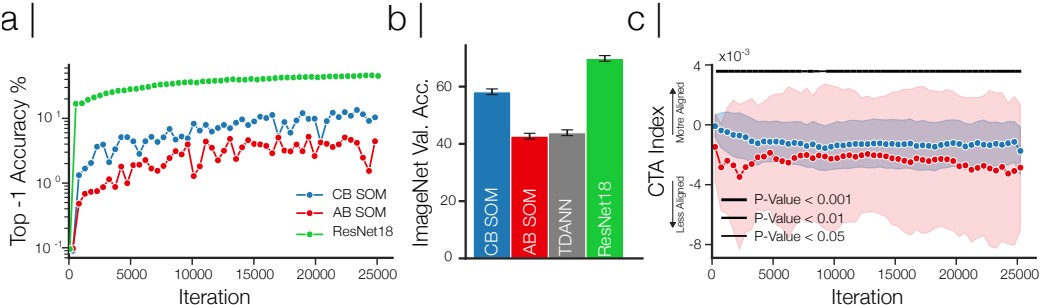

Figure 2: **Credit-based BMU selection aligns topographical and recognition updates**. **a)** ImageNet validation top-1 accuracy during the first few epochs of training for CB-SOM, AB-SOM and ResNet18 models; **b)** ImageNet validation top-1 accuracy for alternative models; **c)** Misalignment between task loss and topographical learning loss (refer to methods) for two models, CB-SOM and AB-SOM, color-coded in blue and red, respectively. Higher numbers indicate greater alignment. The black bar indicates the significance level of the difference in mean between AB-SOM and CB-SOM category-task alignment (CTA) indices at each step.

by a factor of 10 at epochs 60 and 80. The learning rate $\alpha$ were initialized to 0.08 for the first three blocks of ResNet-18, and 0.1 for the subsequent blocks and layers. The neighborhood variance parameter $\sigma$ was initialized to the square root of the number of units. Both $\alpha$ and $\sigma$ were exponentially decayed during training with a factor of -3, although other decay rates were also tested (Figure A4). Importantly, as the BMU selection process is only used during training, both SOM model variations operate similar to the non-topographical ResNet-18 model at test time, with no additional computations. A comparison of training times across models is shown in Figure A5.

## 4 RESULTS

### 4.1 CREDIT-BASED SELF-ORGANIZATION IMPROVES THE ALIGNMENT WITH GRADIENT-BASED REPRESENTATION LEARNING

We first explored whether traditional SOM (AB-SOM) could facilitate learning topographically organized representations within deep networks. We jointly trained the network parameters to both minimize the object recognition loss with gradient descent and induce topography via the AB-SOM local learning rule. We observed that concurrent weight updates with AB-SOM and gradient descent significantly slows down learning of the object recognition task (Figure 2a). Consequently, the AB-SOM model achieved a significantly lower level of performance on the object recognition task compared to the non-topographical model (Figure 2b). We attributed these outcomes to two factors in the AB-SOM algorithm: 1) the bottom-up selection of best matching units (BMUs) based on filter activations, and 2) weight updates designed to better match the observed input. Consequently, *these factors may bias the learning trajectory towards developing filters that reflect the frequency of observed input patterns rather than their utility*. Thus, the network model suffers from a large blow to its object recognition performance.

To quantify this effect, we measured the effect of AB-SOM updates on object recognition loss during training using the category-task alignment (CTA) index that measures the alignment between topography-inducing updates and the object recognition loss (see Extended methods B.1). CTA indices were consistently negative during training of AB-SOM, suggesting that the AB-SOM updates conflicted with gradients necessary for learning the object recognition task (Figure 2c).

We next performed the same analysis on the CB-SOM model in which each BMUs are selected according to their assigned credit (see Extended Methods section). Compared to the AB-SOM model, CB-SOM CTA indices were consistently more positive and closer to 0, indicating a lesser conflict with the gradient-based updates for learning the object recognition task (Figure 2c). In addition, CB-SOM learned substantially faster than AB-SOM model (Figure 2a).

We then compared the object recognition performance of the CB-SOM model against AB-SOM and another recent topographical model, the Topographical Deep Artificial Neural Network (TDANN)

(Margalit et al., 2024) using the ImageNet dataset (Deng et al., 2009). While all topographical models showed reduced performance compared to their non-topographical counterpart with a matched architecture (ResNet-18), the CB-SOM model exhibited a significantly smaller decline in performance (11% for CB-SOM vs. 27% and 26% for AB-SOM and TDANN respectively; Figure 2b). These results empirically confirmed that selecting BMUs based on their assigned credit (i.e. CB-SOM) is significantly more aligned with the inherent dynamics of representations learning via gradient-based algorithms, reducing the conflict that typically accompanies topographical modifications (Figure 2c).

## 4.2 LAYERWISE SELF-ORGANIZATION YIELDS V1-LIKE TOPOGRAPHY IN EARLY MODEL LAYERS

Neurons in the primate's primary visual cortex (V1) are primarily tuned to the orientation, direction of motion, and spatial and temporal frequencies at specific parts of the visual field (Nauhaus et al., 2012). The arrangement of neurons across the cortical surface of V1 follows a systematic organization where 1) neurons with similar selectivity are arranged into "columns" perpendicular to the cortical surface (Hubel & Wiesel, 1962); 2) columns are further arranged such that neurons with more similar selectivity are located at closer parts of the cortex. This structure also gives rise to singularities such as "pinwheels" within which neurons arrange radially around a center point according to their orientation selectivity (Bosking et al., 1997).

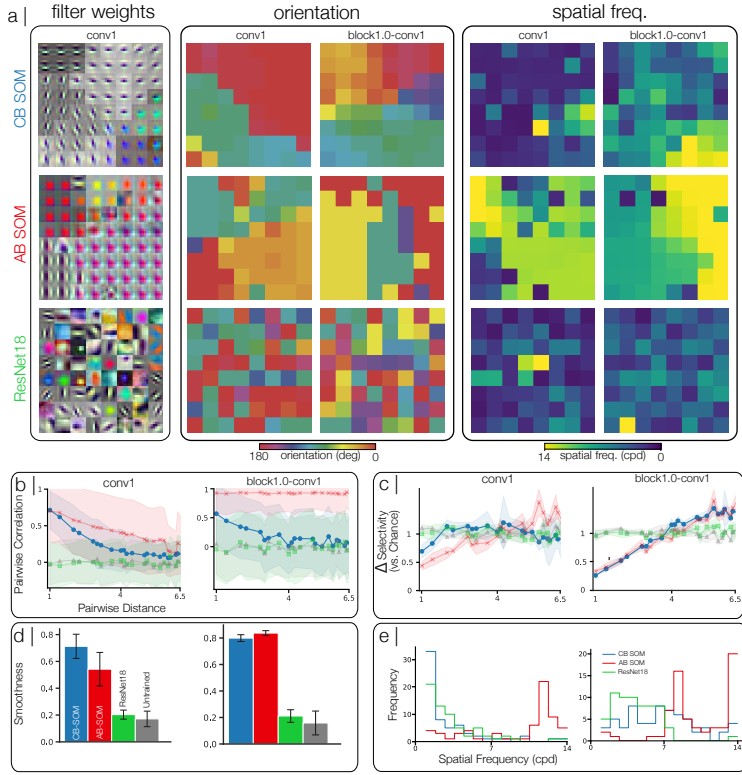

Figure 3: **V1-like organization in early model layers**. **a)** (left) Visualization of the weight parameters in the first layer of each model. Filters in each layer are arranged on a 2D simulated cortical sheet (here 8×8). Orientation (middle) and spatial frequency (right) selectivity is color coded for each filter in two model layers (conv1 and Block.1.0.conv1) for CB-SOM ,ABSOM and ResNet-18. **b)** Pairwise correlations for alternative models as a function of distance on the 2D map; **c)** Change in selectivity as a function of distance on the 2D map, plotted for alternative models and two layers. **d)** Smoothness score for the orientation and of two layers. **e)** The distribution of the spatial frequency selectivity in each layer and each model.

Deep neural networks trained on object categorization task have been shown to develop filters with V1-like properties in their early layers (Cadena et al., 2019). However, the selectivity in these models

does not demonstrate any regularity as a function of position (Figure3a bottom). In other words, the arrangement of these filters appear random when they are placed on a simulated cortical sheet. We first investigated how incorporating self-organizing update rules in training of DNN parameters, affects the arrangement of filters in early layers of these models.

Inspecting the learned parameters of the first layer of the models, we noticed that most first-layer filters in AB-SOM were selective for colored blobs while most filters in CB-SOM consisted of Gabor-like patterns with varying orientations and spatial frequencies (Figure3a top and middle). This provided additional evidence for our guiding hypothesis discussed in the outset, that the bottom-up selection algorithm of AB-SOM biases the learned filters towards more common features in natural images (such as color blobs) rather than more useful ones (such as edges).

We then quantified the filters' selectivity to orientation and spatial frequency in first two convolutional layers of each model (conv1 and block1.0-conv1) using an image dataset of grating patterns of different spatial frequencies (Margalit et al., 2024). In both models, the orientation and spatial frequency selectivity of filters smoothly changed along both simulated physical axes (Figure3a). We computed the pairwise correlations between filter activities as a function of their distance on the simulated cortical sheet which showed a smooth shift in pairwise response correlations as a function of distance for both layers in CB-SOM model but only in AB-SOM's `conv1` layer (Figure3b). Similarly, the difference in orientation selectivity between pairs of filters increased linearly as a function of distance (Figure3c). Both AB-SOM and CB-SOM models demonstrated a significantly higher smoothness in selectivity compared to the non-topographical ResNet-18 model (Figure3d).

Moreover, selectivity in the first and second layers of CB-SOM were reminiscent of the linear zones and pinwheel structure (Blasdel & Salama, 1986; Blasdel, 1992) which have commonly been observed in the retinotopic organization of the visual cortex. In contrast, while the selectivity in the AB-SOM model also transitions smoothly as a function of distance, the emerging organization lacked a clear depiction of signature pinwheel singularity similar to those appearing in the CB-SOM model. In addition, we observed that filters in the AB-SOM model were typically biased toward higher spectrum of spatial frequencies compared to CB-SOM and the non-topographical ResNet-18 architecture that were highly skewed towards lower spatial frequencies (Figure3e). This is important considering the link between selectivity to lower spatial frequencies and object recognition generalization (Li et al., 2023). Similar trends were observed in other layers throughout the model (FigureA2) and an alternative neural network architecture (CORnet-S (Kubilius et al., 2019); Figure A6).

It worth noting that since all hyperparameters were matched between the two models, the higher smoothness of AB-SOM model could not be attributed to the larger neighborhood size in AB-SOM. Further investigation of the source of this difference revealed that the during AB-SOM training, BMUs are consistently selected from a small subset of filters, which leads to over-representation of a few dominant patterns in its layers (Figure A1).

## 4.3 Object-selective clusters emerge in deep CB-SOM layers

Neurons in the Inferotemporal cortex (IT) of human and non-human primates are selective to visual patterns containing particular object categories, most prominently including face, body parts, and scenes (Kanwisher et al., 1997; Epstein & Kanwisher, 1998; Downing et al., 2001). Importantly, neuronal responses in IT cortex are typically invariant to identity-preserving transformations like rotation, scaling, and translation (Rust & DiCarlo, 2010)). Similar to earlier visual areas such as V1, IT neurons are also arranged topographically but instead of orientation and spatial frequency, neurons in this area are organized according to their selectivity to different object categories. While deep neural networks possess a remarkable ability to predict the neuronal response patterns in category-selective brain regions (Ratan Murty et al., 2021b), they do not account for the spatial organization of these category-selective neurons.

We next investigated whether the same organizational principle leads to the formation of filter clusters according to category selectivity as were seen in primate IT cortex. In each model, we examined the organization of category-selectivity across three blocks 2,3,4 (see Methods) and calculated the selectivity of each filter to faces, scenes, body parts, and other non-living objects. Visualizing the resulting selectivity maps for the best IT-like layer in each model on their corresponding 2D physical maps revealed significant clustering of filters according to their category-selectivity in the deep layers of CB-SOM and to a lesser degree in the AB-SOM model (Figure 4a). In particular, category-

selective patches in CB-SOM were more varied, demonstrated stronger category-selectivity (i.e. higher face, body, and place selectivity), and the category-selectivity transitioned more smoothly across the simulated cortical dimensions (Figure 4c; A3).

Moreover, similar to early visual areas, previous studies on non-human primates have also reported an exponentially decaying neural response similarity as a function of distance in the macaque IT cortex (Lee et al., 2020). We observed a similar pattern of decay in pairwise filter response correlations across all layers of the CB-SOM model including the layer which best matched the macaque IT cortex (see Figure A2e). Additionally, similar results were obtained when experimenting with an alternative neural network architecture (CORnet-S (Kubilius et al., 2019); Figure A6).

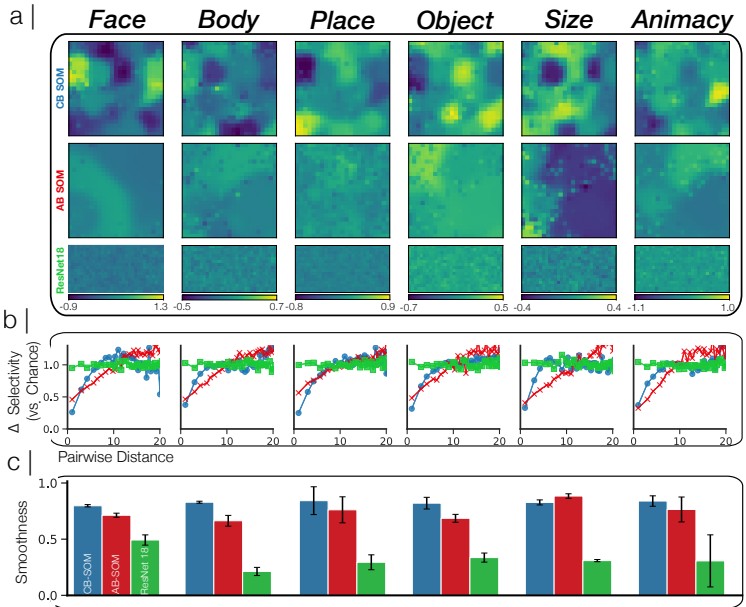

Figure 4: **Category-selective clusters in deep CB-SOM layers. a)** Category-Selective maps for each object category from the fLoc dataset (i.e., Face, Body, Place, and Object), including selectivity for size and animacy. Maps are derived by computing the d-prime value between the target versus other categories; **b)** Pairwise differences in selectivity as a function of distance on the 2D map, analyzed for category selectivity across three models: CB-SOM, AB-SOM, and ResNet-18. **c)** Smoothness measure evaluates the continuity of transition in selectivity within each category-selectivity map and across spatial dimensions of the 2D map. CB-SOM selectivity maps are significantly smoother than both the non-topographical ResNet-18 and the AB-SOM model

## 4.4 IMPROVED BRAIN-LIKE REPRESENTATIONS THROUGH CREDIT-BASED SELF-ORGANIZATION

In the previous sections, we explored the topographical alignment of CB-SOM with that in the human and non-human primates. We next examined whether the filter responses in the CB-SOM model also aligned with the neuronal response patterns across different visual areas. For this, we compared the filter activity in different models with electrophysiological recordings from macaque monkeys as well as fMRI recordings from human subjects.

We began by using the Brain-Score platform (Schrimpf et al., 2018) to quantify the representational similarity between each model and neural activity recorded from various regions of the macaque monkey's ventral visual cortex, as well as to human behavior. Our results indicate that CB-SOM achieved representational similarity scores comparable to those of the non-topographical ResNet-18 model across most benchmarks (Figure 5a), significantly outperforming other baselines such as AB-SOM and TDANN (Margalit et al., 2024). Importantly, the difference between CB-SOM and other topographical models was especially pronounced in the behavioral benchmark, where topographical models typically show a larger gap compared to non-topographical ones. CB-SOM markedly narrowed this gap, highlighting its improved alignment with human behavior.

Next, we used fMRI data from the NSD dataset (Allen et al., 2022) to compare each model's average activity within category-selective clusters (i.e. average response across all filters within those clusters) to the corresponding voxel-averaged activity in human subjects' category-selective patches. For each subject, we utilized their available functional masks to identify the voxels associated with these category-selective areas. We then computed model-brain similarity for each category-selective patch by comparing average responses in the corresponding model patches with the average responses across 3 NSD subjects. The results were compared against alternative topographical models, including ITN (Blauch et al., 2022), TDANN (Margalit et al., 2024), and a non-topographical ResNet-18 model, whose filter activations were linearly combined to predict brain activity in each category-selective patch (Ratan Murty et al., 2021b).

We found that CB-SOM filter clusters predicted fMRI responses in all three human category-selective patches significantly better than all alternative models, even surpassing the ResNet-18 baseline, which used a linear combination of activations to map onto each brain patch (Figure 5b).

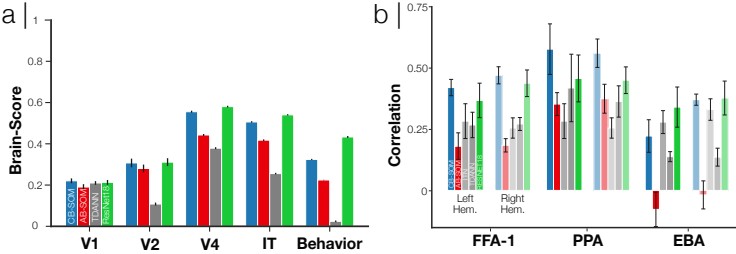

Figure 5: **Representational similarity between models and primates. a)** Assessing representations similarity between models and macaque visual cortex using the BrainScore platform (Schrimpf et al., 2018). **b)** Pearson correlation between best-matching patch in each model and the corresponding cortical patches of Fusiform face area (FFA-1), Parahippocampal place area (PPA), and extrastriate body area (EBA) in each hemisphere using the NSD dataset (Allen et al., 2022). ResNet-18 model was mapped onto each area using cross-validated ridge regression. Mean Pearson correlation is reported for each brain region and error bars indicate the standard deviation across subjects.

## 5 DISCUSSIONS AND LIMITATIONS

In this study, we introduced Credit-Based Self-Organizing Maps (CB-SOM), an adaptation of Kohonen's self-organizing maps that replicates cortical topography in deep convolutional neural networks. Unlike Kohonen's SOM, CB-SOM achieves topographical organization at a much lower cost on feature utility for tasks like object recognition. Our results show that CB-SOM enhances topographical alignment of early and late CNN layers with the early and high-level human ventral visual cortex, respectively, while also improving representational alignment with macaque recordings and human imaging data.

**Bottom-up versus top-down.** CB-SOM, like Kohonen's SOM, is a competitive learning algorithm; however, its key distinction lies in selecting winner units based on their contribution to reducing an ethologically relevant objective function. This top-down approach aligns CB-SOM more closely with gradient descent and significantly influences the network's learned representations.

In Kohonen's SOM, winner units ("Best Matching Units") are selected based on activity levels in response to inputs. As shown in our work, this method often overrepresents frequent but task-irrelevant patterns, leading to less effective representations for tasks like visual object recognition and poorer alignment with macaque and human visual cortex responses. In contrast, CB-SOM's credit-based selection emphasizes task-relevant features, resulting in improved visual recognition performance and stronger representational alignment with the primate brain.

These findings highlight the critical role of task-related top-down signals in shaping topographical organization in the visual cortex and potentially across the neocortex. While this study focuses on visual object recognition, we believe the approach could generalize to other tasks and sensory modalities. Recent studies have shown the feasibility of applying topography-inducing methods

from vision models to language models, where attention and fully connected units in transformer architectures are topographically organized (Binhuraib et al., 2024; Rathi et al., 2024).

**Performance and topography trade-off.** By selecting filters based on the highest gradient of the objective function with respect to parameters, CB-SOM identifies neurons that have the greatest impact on object categorization, prioritizing features like orientation and object selectivity that improve task performance. In contrast, traditional activation-based methods, which select neurons with the highest responses across a batch of images, tend to overrepresent common but less informative features, such as colors (Figure 3). Unlike AB-SOM, CB-SOM emphasizes filters that contribute significantly to improving the objective function, even if they are not highly responsive to images.

We propose that topographical organization in the visual cortex emerges not solely through wiring cost minimization but via self-organizing processes. This is supported by the superior quality of learned representations in CB-SOM, which achieves better representational alignment with neural responses in human and non-human primate brains (Figure 5). These findings suggest that inducing topography in deep networks through overrepresentation of units with maximal task relevance, as in CB-SOM, may yield representations more closely aligned with the brain.

**Towards globally topographic models of visual cortex.** Our approach is distinguished by its ability to achieve universal topography across all layers of the network with minimal impact on task performance. While numerous topographical neural network models have been proposed, most are constrained in the range of brain areas they simulate. Early models of cortical topography were often tailored to mechanisms specific to the primary visual cortex and restricted to that region (Von der Malsburg, 1973; Linsker, 1986). Recent models leveraging deep neural networks, such as those focusing on category-selective neural clusters in later stages of visual processing (Blauch et al., 2022; Doshi & Konkle, 2023; Lee et al., 2020), also exhibit limited applicability. A notable exception is the TDANN model (Margalit et al., 2024), which extended topography-inducing objectives to all network layers, resulting in a universally topographic model. However, this came at the cost of significant performance losses in visual recognition tasks. In contrast, our method provides a biologically-plausible hypothesis for spontaneous topography through self-organization, centered on the idea that Best-Matching Units can be selected via a top-down process—whether by minimizing task error (CB-SOM) or through higher-level cognitive processes like attention.

The universality of CB-SOM topography combined with its high task performance makes it an ideal candidate for predicting the topographical organization of other visual areas, such as V2, V3, and VO. Notably, artificial neural networks are powerful predictors of areas such as V4 (Yamins et al., 2014), enabling neuroscientists to predict neuronal responses and control their activity (Bashivan et al., 2019; Ratan Murty et al., 2021a; Ren & Bashivan, 2024). CB-SOM and other universally topographic neural network models combined with synthesis techniques (Bashivan et al., 2019; Walker et al., 2019; Ponce et al., 2019) provide hypothesis-generation-testing platform for revealing the topographical organization across the visual cortex, similar to recent attempts in rodents (Tong et al., 2023).

**Limitations.** Our work has the following limitations:

- *Lack of formal convergence guarantees.* A key limitation of CB-SOM is the lack of formal theoretical convergence guarantees for its update rule. This challenge is not unique to our work; to our knowledge, no formal proofs of convergence exist for the original SOMs in more than one dimension. CB-SOM introduces a non-local BMU selection mechanism based on a network-wide loss function, which aligns the learning process with top-down objectives. While this innovation improves task performance and representational alignment with biological systems, it adds complexity to proving convergence theoretically. Despite this, all our empirical experiments consistently showed stable training dynamics and smooth topographical organization under standard scheduling of the learning rate and neighborhood function. Future work could explore deriving CB-SOM updates from an explicit energy-based function (Heskes, 1999), to strengthen its theoretical foundation while maintaining its empirical advantages.

- *Remaining performance gap.* Our proposed CB-SOM significantly narrows the performance gap between topographical and non-topographical models in object recognition. However, a noticeable disparity still persists. These findings underscore the potential of task gradients as a promising direction for developing more effective topographical models in the future.

## 6 ACKNOWLEDGEMENTS

This research was supported by the Healthy-Brains-Healthy-Lives startup supplement grant, the NSERC Discovery grant RGPIN-2021-03035, and CIHR Project Grant PJT-191957. P.B. was supported by FRQ-S Research Scholars Junior 1 grant 310924, and the William Dawson Scholar award. All analyses were executed using resources provided by the Digital Research Alliance of Canada (Compute Canada) and funding from Canada Foundation for Innovation project number 42730.

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

# A APPENDIX

## A.1 RELATION TO LATERAL CONNECTIONS IN THE BRAIN

Local lateral connections in the brain are known to connect neurons as an exponentially decaying function of physical distance, a phenomenon thought to be critical for shaping topographical organization. This concept is explicitly embedded in Kohonen's original Self-Organizing Map (SOM) model (Kohonen, 1982), although it is implemented in a modified form via a neighborhood function in later variations (Tuevo Kohonen, 1990). In Kohonen's original formulation (Kohonen, 1982), the activity of each unit $\eta_i$ is computed as follows:

$$\eta_i = \sum_{k \in S_i} \gamma_k \phi\prime_k \tag{5}$$

where $S_i$ denotes the set of neighboring units around the Best Matching Unit (BMU), $\phi\prime$ represents the weighted average of inputs (i.e., the feed-forward computation performed by each unit), and the coefficients $\gamma_k$ are determined based on the distance between each neighboring unit and the BMU. This formulation reflects a form of lateral influence, where increasing the match between the BMU and the input pattern (via a Hebbian learning rule) also leads to updates in neighboring units, with each unit's update weighted according to its distance from the BMU (i.e., modulated by the $\gamma$ coefficients).

In later implementations of SOM, the explicit role of lateral connections in calculating unit activity was often omitted, while their influence on the update rules was retained through the use of a distance-dependent neighborhood function (similar to 3). This simplified formulation retains the impact of lateral connectivity on learning, even though the forward computations no longer directly involves lateral connections.

Our proposed Credit-Based Self-Organizing Map (CB-SOM) utilizes the same update rule as the Kohonen's SOM (Tuevo Kohonen, 1990), preserving this connection to lateral connectivity in the brain. The primary difference lies in how the BMUs are selected; in CB-SOM, BMUs are chosen based on the gradient magnitude, rather than activation levels. This modification aligns CB-SOM with gradient-based learning methods commonly used in deep neural networks, while still incorporating the influence of lateral connectivity principles observed in the brain through the neighborhood function applied during weight updates.

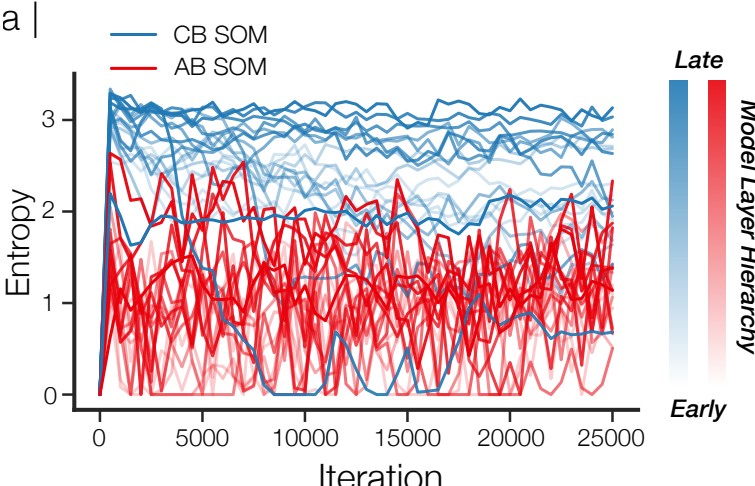

Figure A1: **Diversity of BMUs during training.** We collected the distribution of BMU indices every 500 iterations during training and measured the diversity of selected BMUs by calculating the entropy of BMU distribution for each model and each convolutional layer. Color intensity indicates the depth. Higher entropy corresponds to more diversity in the selected BMUs.

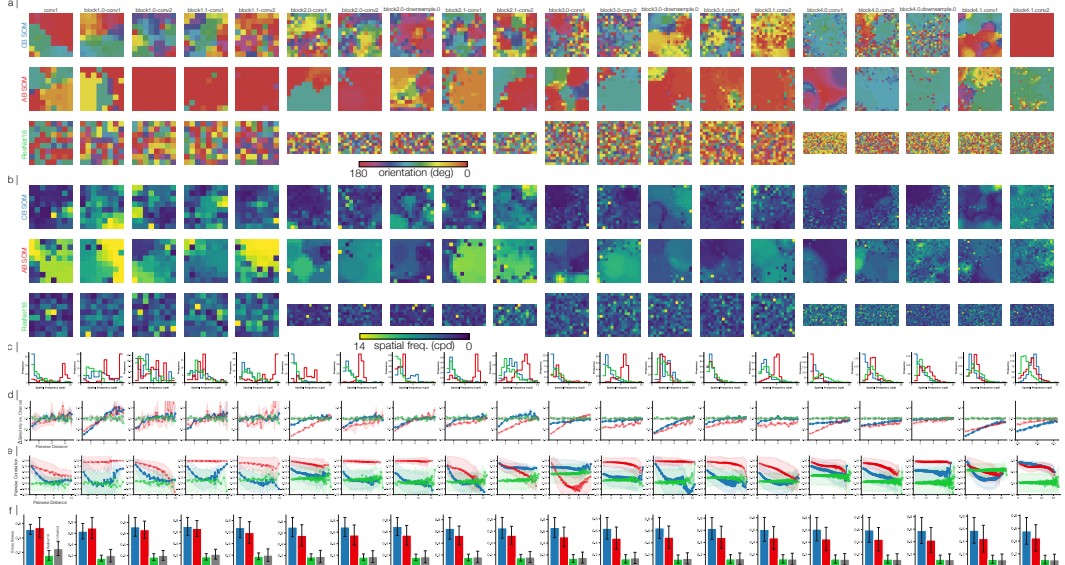

Figure A2: **Visualization of topographical properties for all convolutional layers of CB-SOM , AB-SOM and ResNet-18**. **a**) Orientation selectivity for each filter is calculated and plotted for all convolutional layers. **b**) Similar to **a** but for Spatial Frequency selectivity. **c**) Spatial Frequency selectivity distribution for 3 models plotted for all convolutional layers. **d**) Change in Pairwise orientation selectivity as function of Euclidean distance is plotted for 3 models. **e**) Pairwise Pearson correlation is calculated and plotted as function of Euclidean distance. **f**) Smoothness is calculated for each model from **d** using Eq. 7.

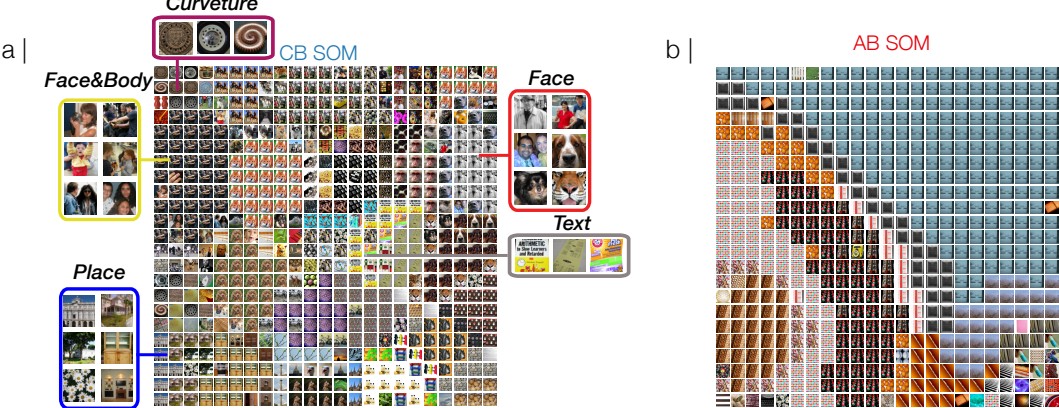

Figure A3: **Most exciting images for the IT layer in each model**. **a**) For each filter in CB-SOM model, the image with highest elicited activation from the ImageNet validation set is plotted on a grid that matches the arrangement of model filters on the 2D simulated cortical sheet. **b**) similar to **a** but for AB-SOM model.

# B    EXTENDED METHODS

## B.1    CALCULATING THE CATEGORY-TASK ALIGNMENT (CTA)

To quantify the impact of SOM updates on model performance in object recognition tasks, we calculated the category-task alignment index (CTA) for each iteration. In order to calculate the CTA index, we calculated the difference in object categorization loss before and after each SOM update. This difference was plotted every 500 iterations.

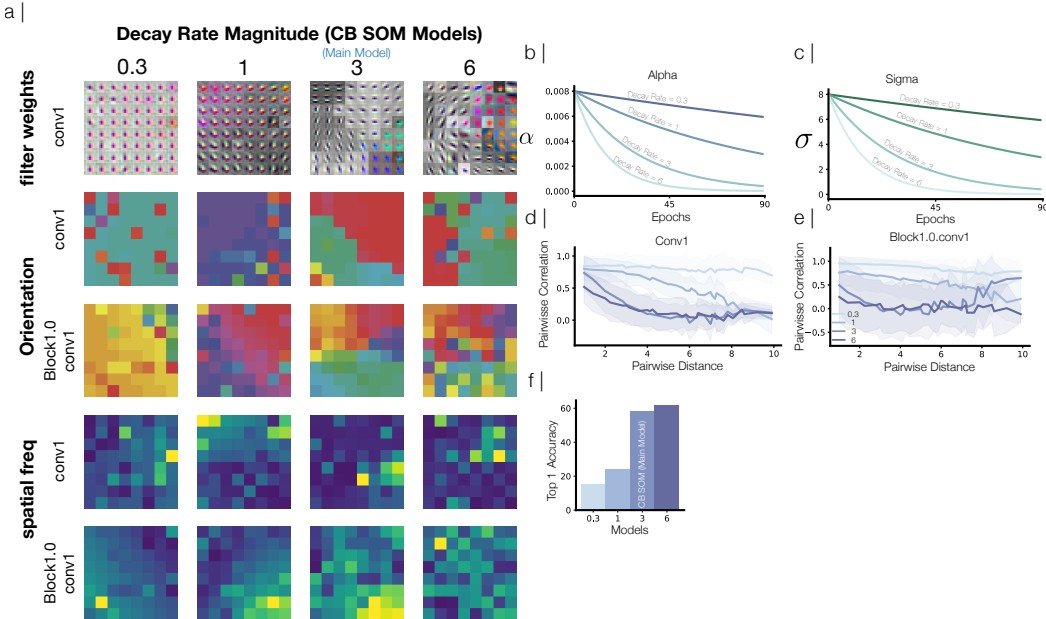

Figure A4: **Training CB-SOM model with various decay rates magnitude**. **a)** visualiation of the first layer filters, orientation selectivity and spatial frequency maps for CB-SOM models trained with different decay rates. **b, c** visualization of reduction of the $\alpha$ and $\sigma$ parameter values during training with different decay rates magnitude. **d, e** Pairwise filter activation correlations as a function of distance on the simulated cortical sheet for models trained with various decay rates. **f** Top-1 ImageNet accuracy for models trained with different decay rates.

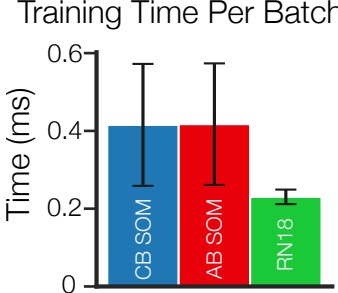

Figure A5: **Training time for CB-SOM, AB-SOM and ResNet18 models per Batch**. Training time for Batch Size 256 for three models. Mean and Standard deviation is calculated across 5000 iterations

$$\text{CTA}_{\text{task}} = \mathcal{L}(\theta) - \mathcal{L}(\theta + \Delta\theta_{\text{SOM}})$$

where $\mathcal{L}$ and $\Delta\theta_{\text{SOM}}$ denote the loss and the SOM update to the model parameters $\theta$ respectively.

## B.2 BMU DISTRIBUTION

We collected the BMU indices every 500 iterations during training and calculated the probability distribution of those indices within each layer. We then quantified the entropy of the BMU distributions for each model and layer. The entropy value captured the spread of the distribution by assigning higher numbers to more uniform distributions with more diverse set of filters selected as BMUs.

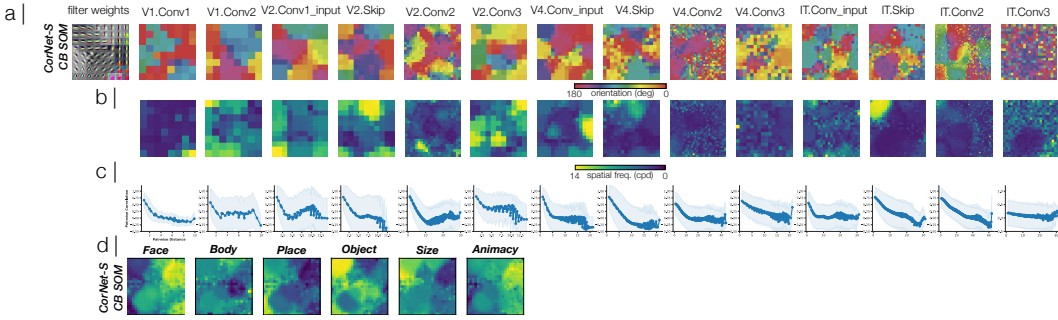

Figure A6: **Generalization of CB-SOM to CORnet-S. a)** Orientation selectivity in convolutional layers of CORnet-s. Each map corresponds to one convolutional layer and each item on the map corresponds to one filter in that layer. Colors correspond to orientation selectivity of different filters in each layer. **b)** Spatial frequency selectivity in layers of CORnet. Colors correspond to the spatial frequency tuning in each filter. **c)** Pairwise correlation as a function of pairwise distance between filters. **d)** Category selectivity for the model's IT layer.

## B.3 V1 ANALYSIS

We assessed the orientation tuning and spatial frequency of filters by using sine grating images from a recent publication (Margalit et al., 2024). The imageset contained eight orientations, ranging from 0 to 180 degrees and eight spatial frequencies from 0.5 to 12 cycles per degree. Using a similar approach to this work, we assessed the orientation and spatial frequency by constructing the tuning curves and quantifying the strength of the tuning curves using circular variance (CV):

$$CV = 1 - \frac{\left|\sum_k r_k e^{i2\theta_k}\right|}{\sum_k r_k} \tag{6}$$

where $\theta_k$ is the $k$-th orientation (in radians) and $r_k$ is the response of the filter to that orientation. The orientation tuning curves are additionally fit with a von Mises function, and the peak of this function is taken as the preferred orientation.

For plots in Figure 3c ($\Delta$Selectivity), we assessed the change in selectivity tuning ($\Delta$Selectivity by calculating the difference in orientation selectivity as a function of pairwise distance between the filters. Similarly, for plots in Figure 3d (Smoothness), we assessed the smoothness of orientation and spatial frequency maps in each model using the approach from Margalit et al. (2023).

1. We constructed a vector of pairwise filter activation correlations against their cortical distance for each layer. This vector represents the correlation of responses between model filters based on their spatial proximity in the cortical structure.

2. Using this vector, we quantified the smoothness of the selectivity maps by comparing the selectivity of the closest model filter pairs with those of the least similar pairs. This allows us to assess how tuning similarities vary with cortical distance.

3. Let $x$ represent a vector of pairwise tuning similarity values arranged in ascending order of cortical distance. The smoothness score $S(x)$ is defined as:

$$S(x) = \frac{\max(x) - x_0}{x_0} \tag{7}$$

where $\max(x)$ is the maximum similarity value among the closest pairs, and $x_0$ is the minimum similarity value among the least similar pairs.

4. A higher smoothness score indicates that closely located filters exhibit more similar functional characteristics, suggesting a smoother transition in the selectivity maps. Conversely, a lower score reflects greater variability in selectivity, implying abrupt changes as the distance increases.

### B.4 Finding category-selective patches

We used the fLoc datasets (Stigliani et al., 2015) to find the category-selective patches in models containing four image categories: Face, Body, Place, and Objects, where there are 144 images per category. We calculated the sensitivity of each filter for each category by measuring the difference between the mean responses of each category's images versus all other categories.

$$d' = \frac{\mu_c - \mu_o}{\sqrt{\frac{\sigma_c^2 + \sigma_o^2}{2}}} \tag{8}$$

where $c$ and $o$ denote the target and other categories respectively.

### B.5 Training CORnet with CB-SOM

We demonstrate the applicability of CB-SOM to training an alternative architecture by applying it to CORnet-S across all convolutional layers (Kubilius et al., 2019). For the CorNet-S architecture, an initial alpha of 0.08 was used for the V1 and V2 blocks, while an alpha of 0.1 was selected for the V4 and IT blocks. The initial sigma parameter was determined based on the square root of the number of units in each layer.

### B.6 Aligning neural network model layers and brain areas

We quantified neural network model and brain alignment in two ways:

1. **BrainScore similarity.** We used the publicly available BrainScore platform (Schrimpf et al., 2018) to measure the similarity of each models with neurophysiological and behavioral data. Briefly, BrainScore uses principal component regression to find combination of units (i.e. filter activations) in each layer of the model that can predict the responses in recorded neural sites. The normalized image-level behavioral consistency measure (I2n) (Rajalingham et al., 2018) is used to assess the behavioral consistency with human subjects.

2. **Model-brain patch similarity**. We analyzed the human fMRI data from Natural Scene Dataset (Allen et al., 2022), which contains the fMRI responses for various cortical regions including, the Fusiform face area (FFA), Parahippocampal place area (PPA), and extrastriate body area (EBA) of 8 subjects to a total of 73,000 natural images. After identifying the category-selective model patches using the fLoc stimulus set (Stigliani et al., 2015), we collected the 2D coordinates of filters that were selective for faces, body parts, and places. We then used the density-based spatial clustering algorithm (DBSCAN)(Ester et al., 1996) to label filters into defined patches. We considered a filter cluster as a patch if it contained at least three filters with a physical distance of one. For the TDANN model, we used the method described in (Margalit et al., 2023) to identify the category-selective patches. We then averaged the responses of all neurons within each cluster and computed the Pearson correlation between the model patch activity and the corresponding cortical patch. All comparisons were performed using 2-fold cross-validation where correlations on the train set were used to select the best layer and patch in each model, and only the test values were reported. For ResNet-18, we performed ridge regression on the activation of the model to the brain areas. We then performed cross-validation five times, and the average correlation and standard division for the testing set were reported. For each subject, we calculated the correlation for each brain area across all models using the method described above. We then reported the average testing correlation and standard deviation across subjects for all models.

We identified layers to be IT-like by measuring the correlation between category-selective patches in model and brain (see Model to Brain patch similarity above), and selecting layer with highest correlation.

