# OpenReview forum: "Credit-based self organizing maps: training deep topographic networks with minimal performance degradation"
_ICLR.cc/2025/Conference — ICLR 2025 Poster_

### Official Review · Reviewer_aFS5 · 2024-10-26

**Soundness:** 3
**Presentation:** 3
**Contribution:** 1
**Rating:** 5
**Confidence:** 4

**Summary:**

The paper contributes an original idea for training supervised deep neural networks models whose layers show topographical organization due to self-organization. It is shown that object recognition performance is reduced less by this than by a naive approach of introducing self-organization. It is further shown that DNNs trained by this CB-SOM algorithm exhibit orientation selectivity in lower layers and category selectivity in higher ones. Comparisons to human FMRI studies are performed, indicating as good match.

**Strengths:**

The paper presents an original idea to tackle the long-standing problem of generative classification. It is well-organized and easy to read, with some minor glitches here and there. The technical quality of conducted experiments is good. It shows that self-organization can be shown when training CNNs on complex benchmarks like ImageNet while degrading classification performance significantly less than other models.

**Weaknesses:**

The main weak point is that the advantage of topographical organization is somehow assumed to be self-evident. Is there a functional advantage to having this property? As long as this is unclear, it is hard to accept that performance degradations as you show them are acceptable

Another issue is that the CB-SOM update rule is derived ad hoc, without reference to minimizing a loss. So there are no convergence guarantees or anything, which is bad from a conceptual perspective. It would be strongly desirable to derive this update rule from a loss function that is added to the supervised loss. A possibility is the energy-based SOM model of Heskes [1]

Furthermore, some technical aspects of the model are not clearly described, see comments below

Lastly, orientation/category-selectivity must be compared to a purely supervised model to assess whether the effect would not have occurred without CB-SOM. It is well-known that filters in lower CNN layers develop orientation selectivity as well.

[1] Heskes, Tom. "Energy functions for self-organizing maps." Kohonen maps. Elsevier Science BV, 1999. 303-315.

Comments:
Related work could include works on topologically organized GMMs [2,3] as well as energy-based SOM [1].
3.1 can be more concise, the formula is well known
3.2 is unclear. What are "the errors between each layer’s winning unit’s weight and other units’ weights within that layer ..." ? Just the same as $w_c(t)-w_{ij}(t)$ for AB-SOM? A formula would help here!
3.2 should be derived from minimizing a loss, e.g., Heskes? Otherwise, what is justification for post-hoc weight adjustment?
3.3 How does a 3D structure (neurons are organized in width/height/channel dimensions) like a convLayer fit with the 2D organization assumed in CB-SOM? This must be explained as it is critical to the understanding of the approach
4.1 It should be better described how training AB-SOM works in this setting
4.1 from line 243: CTA is unclear. Either explain fully, here, or not at all and leave to appendix
Fig.3a: what are we seeing in the orientation-selectivity plots? All filters arranged on a 2x2 grid? Not terribly clear to me...
All of Sec.4: this analogy to human brains is interesting, but does it serve a purpose? Why would we want this, given that object recognition is impaired by it? This question needs to be answered, imho, to assess the value of these contributions
All of section 4: The observation of category/orientation selectivity would more compelling if you compared these to plots for a ResNet that was trained in supervised fashion only, without the CB-SOM "add-on".

**Questions:**

The observation of category/orientation selectivity would more compelling if you compared these to plots for a ResNet that was trained in supervised fashion only, without the CB-SOM "add-on". Because it is not clear that the selectivity you observe does not come from the supervised part already. Do you have any insights on this?

---

> ### Author Response · Authors · 2024-11-14
> **missing references [2,3]**
>
> Dear reviewer,
>
> Thank you for your comments and suggestions. We are actively working on preparing a response to your comments. However, we noticed that the information about references 2 and 3 in your comments (related to topologically organized GMMs) may be missing or can't be viewed by us. We would appreciate it if you could share the information about these references with us.
>
> Authors

---

> > ### Author Response · Authors · 2024-11-21
> > **Response to initial comments 1/2**
> >
> > We thank the reviewer for acknowledging the originality of our approach and the significance of tackling the challenge of introducing self-organization into deep neural networks. We appreciate their recognition of the technical quality of our experiments and the clarity of the manuscript. Below: we have carefully addressed the reviewer’s concerns and refined the manuscript to further highlight the contributions and implications of our work.
> >
> > **Functional relevance of topographical organization.**
> > We understand the reviewer’s concern and are happy to provide further context to help the reviewer appreciate the contributions of our work.
> >
> > While the main motivation of our research is rooted in neuroscience, we believe that the resulting topographical models have potential advantages for both neuroscience and AI.
> >
> > From the neuroscience perspective, while the true functional role of cortical topography is still being debated, there are several theories about their function:
> >
> > 1) The current most widely-accepted theory is wiring cost minimization arguing that the main evolutionary function of the topographical organization in the brain has been to reduce the amount of nerves connecting different cortical circuits together which is deeply important for maintaining the physical size of the neural circuitry while boosting its capacity [1].
> > 2) A recent work has also suggested a possible role for the cortical topography in learning robust representations when considering the lateral connections directly in the architecture of the neural network model [2]. However, neither AB- or CB-SOM showed any indication of improved robustness beyond the non-topographical model.
> >
> > Regardless of their functional roles, topographical neural network models constitute immensely useful tools for guiding therapeutic interventions such as stimulation or surgeries [3]. For example, planning resection surgeries in patients with epilepsy could benefit from predictions from topographical models that match the organization of the cortex as surgeons often plan their surgeries to avoid cortical patches that serve critical functions such as face recognition.
> >
> > From an AI perspective, inducing local structure in deep networks may potentially enhance interpretability of models and possibly improve our ability to investigate and reduce harmful behaviors via ablating local clusters in models that are closely associated with undesired behaviors.
> >
> > We update the introduction and discussion section to better incorporate the motivation as well as potential benefits of our research (lines 50-58, 515-522).
> >
> > [1] Jacobs, R. A., & Jordan, M. I. (1992). Computational consequences of a bias toward short connections. Journal of cognitive neuroscience, 4(4), 323-336.
> >
> > [2] Qian, X., Dehghani, A. O., Farahani, A., & Bashivan, P. (2024). Local lateral connectivity is sufficient for replicating cortex-like topographical organization in deep neural networks. bioRxiv, 2024-08.
> >
> > [3] Schrimpf, M., McGrath, P., Margalit, E., & DiCarlo, J. J. (2024). Do Topographic Deep ANN Models of the Primate Ventral Stream Predict the Perceptual Effects of Direct IT Cortical Interventions?. bioRxiv, 2024-01.
> >
> > **Lack of formal convergence guarantees.**
> > We appreciate the reviewer’s insightful comment. While it is true that CB-SOM, like many self-organizing map (SOM) variations, does not include such guarantees, this limitation aligns with the broader landscape of SOM research. To our knowledge, no formal proof of convergence exists for the original SOMs in more than one dimension. Much of the progress in this area has been empirical, relying on simulations rather than theoretical guarantees.
> >
> > The CB-SOM algorithm introduces a key modification by selecting BMUs based on a network-wide loss function, rather than local unit activity. This adjustment aligns the BMU selection process with top-down learning objectives, which contributes to CB-SOM’s superior performance in object recognition tasks. However, this non-local mechanism complicates the formalization of an energy-based function or direct proof of convergence, adding to the challenges already present in proving convergence for standard SOMs.
> >
> > While we acknowledge this as a limitation, we emphasize that our extensive empirical results consistently demonstrate stable training dynamics, smooth topography, and reproducible outcomes across all experiments. Following standard practices, we applied typical scheduling for the learning rate and neighborhood function, and we have not encountered any instances of instability or failure to converge.
> >
> > In response to the reviewer’s suggestion, we have highlighted this point as a limitation in the revised “Discussion, Limitations, and Conclusions” section. Furthermore, we acknowledge the relevance of energy-based models such as the one proposed by Heskes (1999) and agree that exploring such approaches could be a promising future work (lines 526-535).
> >
> > (Continues in the next comment)

---

> > > ### Author Response · Authors · 2024-11-21
> > > **Response to initial comments 2/2**
> > >
> > > (Continued from previous comment)
> > >
> > > **Orientation/category-selectivity in topographical and non-topographical models.**
> > > We would like to first clarify that we do not claim that orientation and category-selectivity strictly emerges in topographical models and are missing in non-topographical ones. As the reviewer also pointed out, both forms of selectivity emerge in non-topographical models as well. The main agenda concerning topographical neural network models is regarding the distribution of selectivity across a 2-dimensional simulated cortical sheet over which the channels/filters are arranged (as depicted in the Fig. 1).
> > >
> > > In our original submission, we compared the orientation selectivity between the two topographical models and the non-topographical ResNet18 in Fig. 3a which shows that the channels in ResNet18 model also show diverse selectivity to different orientations but that their arrangement across the 2D simulated cortical sheet is random. In the revised manuscript, we additionally updated Fig. 4 and included plots for the non-topographical ResNet18 as well to showcase that category-selective units also exist in the non-topographical model although not topographically organized.
> > >
> > > We hope these changes have addressed the reviewer’s concern and remain open to further suggestions.
> > >
> > > **Missing related work.**
> > > As posted in an earlier comment, we were only able to see the information for the Heskes paper and not the other two references to topographically organized GMMs. As discussed above in response to “Lack of formal convergence guarantees”, we have carefully considered the Heskes paper and its relevance to your present work. *We would be happy to consider and discuss the other two references suggested by the reviewer if the reviewer would kindly share them with us*.
> > >
> > > **More concisely written section 3.1.**
> > > We thank the reviewer for the suggestion and revised the text in section 3.1 to address the reviewer’s concern.
> > >
> > > **Clarifying section 3.2.**
> > > We thank the reviewer for their comment. We revised the text in section 3.2 to improve clarity and readability. Specifically, in addition to revising the text, we also added a reference to the update rule used for updating the parameters, which is similar to AB-SOM.
> > >
> > > **Justification for post-hoc weight adjustment in CB-SOM.**
> > > As we discussed above, the weight update rule in Kohonen’s SOM model is primarily driven by neuroscientific observations. CB-SOM applies the same weight update rule as that in AB-SOM and Kohonen’s SOM. The update rule is primarily derived from observed characteristics of the local connectivity between neurons in the visual cortex that follow an exponentially decaying pattern as a function of distance. The neighborhood function in Eq. 3 approximates the effect of the functional interaction between nearby cortical neurons on the Hebbian-like weight updates (Kohonen 1990). We updated the text in section 3 to clarify that the neighborhood function is derived from Kohonen 1990 and its relation to lateral feedback connections (lines 181-184 and Appendix A.1).
> > >
> > > **Mapping the 3D structure of activations to a 2D sheet.**
> > > We agree with the reviewer that this point may have not been adequately clarified in the manuscript. Briefly, we consider each filter as a unit on the 2D simulated cortical sheet. The activity corresponding to that unit/filter/channel is considered as the activity of that filter averaged across the spatial dimensions (HxW). To address the reviewer’s question and clarify this point, we updated Fig. 1 to and revised the text in section 3 and highlighted that individual filters are arranged on the 2D simulated cortical sheet (lines 164-167).
> > >
> > > **Unclarity of CTA in section 4.1.**
> > > Due to space limitations, we chose to include the definition of the CTA index in the appendix and only referred to what the values indicate in text. To address the reviewer’s concern we further clarified what the CTA constitutes in the main text, made adjustments to Fig. 2c and revised the associated text in Extended Methods to improve readability (lines 260-261, 795-802).
> > >
> > > **Clarifying what’s shown in Fig 3A.**
> > > The arrangement of filters on the orientation selectivity plot follows the same order as in the other two subplots (filter weights and spatial frequency). Essentially, all 64 filters in each of the two layers are arranged on a 8x8 matrix. Each item on the matrix is associated with one of the filters in the corresponding layer. The associated colors indicate the orientation selectivity of each of the filters as shown in the colorbar below the subplots. We revised the figure caption to include a more detailed description of what this plot shows.

---

> > > > ### Comment · Reviewer_aFS5 · 2024-11-26
> > > >
> > > > Deear authors,
> > > >
> > > > thank you for you clarifications and updates in the paper. Some of my concerns have been adressed, as far as it concerns clarity and completeness.
> > > > However, my concerns remain w.r.t. following issues:
> > > > - lack of a proper loss function for your model. I still believe this is a major problem, as your ad-hoc learning rule may fail to lead to a stable learning result for other problems. True, this is the same case for SOMs, but there at least convergence has been established through two decades of experimentation, and indeed it can be shown that SOM approximates the Heskes loss function. At a conference about deep learning, I think this is a serious flaw in your model.
> > > > - gains -vs- cost. The neuroscience inspiration is interesting but it fails to provide tangible benefits. And I do not believe for a second that brain surgeons are guided by SOM-based models when performing interventions, however interesting the analogy may be. Indeed, there are higher brain areas where no obvious topographic organization is observed. At the same time, your modification leads to a significant performance drop that cannot be compensated by a bigger DNN, or else you would have shown this. So all in all, there are no tangible gains but a pronounced performance loss, which is problematic.
> > > >
> > > > I will raise my score by one point, acknowledging your clarifications.

---

> > > > > ### Author Response · Authors · 2024-11-26
> > > > > **thank you!**
> > > > >
> > > > > We sincerely thank the reviewer for raising their score and for engaging deeply with our work. We acknowledge that some potential benefits of topographical neural network models, such as their application in surgical interventions, may currently seem speculative. However, as these models advance and integrate more biological and functional realism, we believe they could become practical tools in neuroscience and medicine in the near future. From an AI perspective, we maintain that inducing local structure in deep networks offers promise for enhancing model interpretability, potentially aiding in identifying and mitigating undesired behaviors by targeting specific clusters associated with such outcomes. We appreciate the reviewer’s insights and hope our study inspires further exploration into both the neuroscience and AI applications of topographical models.

---

### Official Review · Reviewer_nVAR · 2024-10-28

**Soundness:** 3
**Presentation:** 4
**Contribution:** 3
**Rating:** 8
**Confidence:** 5

**Summary:**

This paper proposed a method for seamlessly incorporating topographical learning with top-down supervised learning.
This paper's primary contribution is the proposal of an Action-Based Self-Organizing Map (CB-SOM) that utilizes the gradient of a loss function as the criterion for selecting the Best Matching Unit (BMU) for topographical learning.

The paper is generally very well written and supported with interesting and valid experiments.

The clarity of the paper was significantly improved after the revision.

**Strengths:**

The proposed method makes sense, mainly when a top-down loss function can be defined, in that the neural network should select a BMU based on the contribution of the topographical hidden neurons to the loss function. The reviewer agrees with the authors that the integration of the conventional SOM (AB-SOM) suffers from over-representations of some hidden neurons that do not necessarily contribute to the learning performance of the whole network. The proposal clearly alleviates this discrepancy.

This paper also offers interesting arguments that compare the resulting CB-based SOM with primates, which is essential to argue about the biological validity of the proposed model.

The paper is also very clearly written and easy to understand.

**Weaknesses:**

There are no essential weaknesses in this paper. But there are some unclarities, listed in the Questions parts of this review, that should be clarified before this paper is ready for publication.

**Questions:**

1. As the selection of a BMU always depends on the gradient of a loss function, it is unclear how the network chooses a BMU without a loss function, for example, in the neural network's running stage. It is outside the scope of the paper, but biological neural networks also deal with reinforcement learning. Hence, the authors need to give at least their comments on how to deal with this matter.

2. It is unclear whether a reconciliation between AB-SOM and CB-SOM occurs, at least in the latter stage of the learning rule. In short, does the CB competition converge into the AB competition in the latter stage of the learning process? It would be insightful if the authors could prove or disprove that CB competition is the transient state of AB competition.

3. There is a model that seamlessly combines topographical learning and top-down reinforcement learning as follows:
a) P. Hartono, P. Hollensen and T. Trappenberg, "Learning-Regulated Context Relevant Topographical Map," in IEEE Transactions on Neural Networks and Learning Systems, vol. 26, no. 10, pp. 2323-2335, Oct. 2015, doi: 10.1109/TNNLS.2014.2379275.
b) P. Hartono, "Mixing Autoencoder With Classifier: Conceptual Data Visualization," in IEEE Access, vol. 8, pp. 105301-105310, 2020, doi: 10.1109/ACCESS.2020.2999155.

These two papers share many similarities with the proposed paper. They also derive a new modification rule for the reference vectors associated with the topographical neurons. They demonstrate that the conventional SOM may be a particular case for general topographical representations in supervised models.

The authors must argue their proposal's novelties and advantages compared with these past works.

---

> ### Author Response · Authors · 2024-11-21
> **Response to initial comments**
>
> We sincerely thank the reviewer for their thoughtful feedback and for recognizing the effectiveness of our proposed approach and for acknowledging the clarity of our writing and the validity of our experiments. In response to the reviewer’s comments, we have further refined our arguments and addressed all concerns to strengthen the manuscript.
>
> **Inference without a loss function.**
> The BMU selection process is only used at training time to induce a topographical organization across the model filters. However, during inference time the model simply computes the output of all filters without the need for calculating the BMU unit and in that sense operates similar to a non-topographical network. For that reason, the network doesn’t require a particular loss or reward function to operate after it is trained. We clarified this point in section 3 (Model Training) of the revised submission (lines 237-239).
>
> **Reconcilitation between AB- and CB-SOM.**
> We were unsure whether we understood the reviewer’s point correctly but we considered two possible interpretations:
> We assumed that by “reconciliation between AB-SOM and CB-SOM occurs” the reviewer hypothesizes that the two models AB- and CB-SOM eventually converge to the same representation after sufficient training. Several of our experimental results support the view that such reconciliation does not occur in these models: 1) The visualization of weights in the first convolutional layer in Fig 3a shows drastic qualitative differences in the type of filters emerging in the first layer of the network that would very likely have substantial effect on the type of representations emerging in later layers of the network; 2) Related to the previous point, the distribution of frequency selectivity in the two models are substantially different between the two models as shown in Fig. 3e; 3) Unlike in CB-SOM, units in deep layers of AB-SOM have significantly lower selectivity to categories such as Face, Body, and Place; 3) The Representational similarity to macaque and human visual cortices is quite different between the two models judging by the results in Fig. 5; 4) Finally the performance of the two models on object recognition is substantially different according to Fig. 2b.
>
> Alternatively, the reviewer may have suggested that while the representations are different, the probability of units being selected as BMUs may converge to similar values in the models. As shown in Fig. A1 in the appendix, we showed that this point is also not true as the entropy of the distributions are quite different between the two models with CB-SOM having a higher entropy (more uniform likelihood for units being selected as BMUs) compared to AB-SOM.
>
> We hope our explanation adequately answers the reviewer’s question and we remain open to clarifying further.
>
> **Related work by Hartono et al.**
> Thank you for bringing these two highly-related works to our attention. In both works, topographically organized feature maps are learned via the context-relevant SOM approach which combines the SOM learning rule (including both BMU selection and update rule) with backpropagation of error given a loss function. In that sense, we believe that this method is effectively similar to the AB-SOM baseline considered in our work that combines the original SOM update rule with backpropagation using the cross-entropy loss. We included and discussed both references in the Related Work section of the revised submission (lines 99-101).

---

> ### Comment · Reviewer_nVAR · 2024-11-26
> **comment to the authors' responses (Nov. 26)**
>
> I appreciate your response in such a short time.
> You have clarified all of my questions very well. This is a good paper.
>
> I only have some minor issues that need further clarifications and modifications.
>
> 1. There is no neighborhood function during the operation stage. All of the filters generate their outputs without any topological constraints. Is my understanding correct?
> 2. Regarding the difference between your model and the rRBF in [Hartono 2015, 2020]: while their model did not explicitly select the BMU based on the loss gradient, the loss gradient is included in the filters' modification. Hence, the filters with large gradient losses are the ones that are modified the most. In contrast, filters with low gradient values (credits) will be less modified. Doesn't this make your model practically equivalent to theirs?
> 3. Line 634 exceeds the frame of the paper.
> 4. [Hartono 2015, 2020] are journal papers, not conference papers, so please fix line 596 and line 602,

---

> > ### Author Response · Authors · 2024-11-26
> > **further clarifications**
> >
> > We appreciate the reviewer's continuing encouragement and support. See below for responses to your questions
> >
> > 1. That is correct. To clarify further, the topological constraints are only enforced during network training using the SOM update rule. After training is done, this topographical organization is already embedded in the network weights (i.e. filter weights arranged on the 2D simulated cortical sheet) and the constraints are no longer needed to make predictions.
> >
> > 2. As mentioned in our previous response, the method described in [Hartono 2015, 2020] consists of an SOM update rule guided by the bottom-up generate unit activities in combination with gradient-based learning using an objective. This procedure is the same in the AB-SOM model considered in our work as well where the SOM was guided by the unit activations in a similar manner and in combination with the gradient based learning. The hypothesis put forth by the reviewer "...the filters with large gradient losses are the ones that are modified the most. In contrast, filters with low gradient values (credits) will be less modified." is indeed reasonable. However, our results demonstrate that using a gradient-guided competition in the SOM algorithm, as implemented in CB-SOM, leads to substantially different and improved outcomes compared to AB-SOM. CB-SOM significantly outperforms AB-SOM across all metrics, underscoring the practical advantages of our approach.
> >
> > 3,4. Thank you for the comments. We have corrected these issues in the revised submission.

---

### Official Review · Reviewer_WLW9 · 2024-11-01

**Soundness:** 3
**Presentation:** 3
**Contribution:** 2
**Rating:** 6
**Confidence:** 4

**Summary:**

This study proposes an intriguing way to force topography in ResNet18. Earlier topographical models had exhibited significant drops in accuracy, but this study provides an effective way to alleviate the accuracy drop.

**Strengths:**

The authors extensively compared the model’s responses and neural responses recorded from Nonhuman primates and humans, which will greatly interest readers of brain science research. Their study may also be of interest to the deep learning research community because it proposes a way to impose local structure to deep learning models, which may aid us build domain-specific models with specific local structure. The authors evaluated a single architecture only. It would be more compelling if they tested more architectures.

**Weaknesses:**

The authors evaluated a single architecture only. It would be more compelling if they tested more architectures.

**Questions:**

I do have two questions.

When the authors mention “units” of ResNet18, are they referring to ResNet18’s channels? If so, some clarification is needed. If not, a detailed explanation is needed.

In general, lateral connections in the brain have been believed to play a role in shaping topography. My second question is, could the authors provide insights into how their proposed mechanisms are related to lateral connections in the brain?

---

> ### Author Response · Authors · 2024-11-21
> **Response to initial comments**
>
> We thank the reviewer for recognizing the significance of our work and its potential interest to both neuroscience and deep learning communities, as well as for acknowledging our extensive comparisons between model responses and neural recordings. Below we have carefully addressed each of the reviewer’s concerns and questions to further enhance the manuscript.
>
> **Evaluating alternative architectures.**
> We thank the reviewer for their helpful suggestion. We agree with the reviewer that showcasing the generality of the proposed algorithm is critical. To address the reviewer’s comment, we trained an alternative neural network architecture (CORnet-S [1]) following similar settings as those used for ResNet18 architecture. We found strong topographical organization in the early and late layers of this network, mimicking those we had obtained in our ResNet18 experiments. We added these new results to the new Fig. A6 in the appendix.
>
> [1] Kubilius, Jonas, Martin Schrimpf, Kohitij Kar, Rishi Rajalingham, Ha Hong, Najib Majaj, Elias Issa et al. "Brain-like object recognition with high-performing shallow recurrent ANNs." Advances in neural information processing systems 32 (2019).
>
> **Clarification of “units”.**
> We thank the reviewer for bringing this issue to our attention. We agree that the language used in our original manuscript was not completely clear. Indeed, by “units” we refer to channels in each convolutional layer. We revised Figure 1 and text in section 3 and clarified this point by replacing “unit” with “filter” throughout.
>
> **Relation to lateral connections.**
> We appreciate the reviewer’s interest in understanding the relationship between our proposed mechanism and lateral connections in the brain. Local lateral connections in the brain are indeed known to connect neurons as an exponentially decaying function of physical distance, which helps to shape topographical organization. This concept is explicitly embedded in Kohonen’s original Self-Organizing Map (SOM) model (Kohonen, 1982), although it is implemented in a modified form in later variations.
>
> Briefly, in Kohonen’s original formulation, the activity of each unit $\eta_i$ is computed as follows:
>
> $\eta_i = \sum_{k\in S_i} \gamma_k \phi{\prime}_k$
>
> where $S_i$ denotes the set of neighboring units around the Best Matching Unit (BMU), $\phi{\prime}$ represents the weighted average of inputs (i.e., the feed-forward computation performed by each unit), and the coefficients $\gamma_k$ are determined based on the distance between each neighboring unit and the BMU. This formulation reflects a form of lateral influence, where increasing the match between the BMU and the input pattern (via a Hebbian learning rule) also leads to updates in neighboring units, with each unit’s update weighted according to its distance from the BMU (i.e., modulated by the $\gamma$ coefficients).
>
> In later implementations of SOM, the explicit role of lateral connections in calculating unit activity was often omitted, while their influence on the update rules was retained through the use of a distance-dependent neighborhood function. This simplified formulation retains the impact of lateral connectivity on learning, even though the computation of activity no longer directly involves lateral connections.
>
> Our proposed Credit-Based Self-Organizing Map (CB-SOM) utilizes the same update rule as the classic SOM, preserving this connection to lateral connectivity in the brain. The primary difference lies in how the BMUs are selected; in CB-SOM, BMUs are chosen based on the gradient magnitude, rather than activation levels. This modification aligns CB-SOM with gradient-based learning methods commonly used in deep neural networks, while still incorporating the influence of lateral connectivity principles observed in the brain through the neighborhood function applied during weight updates.
>
> We included the above information about the relation between lateral connections and the SOM models in a new appendix section A.1 of the new submission.

---

> > ### Comment · Reviewer_WLW9 · 2024-11-26
> >
> > I appreciate the authors’ response, but I will keep my previous rating.

---

> > > ### Author Response · Authors · 2024-11-26
> > > **Thank you**
> > >
> > > We appreciate the reviewer’s thoughtful feedback and the opportunity to improve our manuscript. The additional experiments, clarifications, and extended discussion in the revised submission were aimed at addressing the reviewer’s concerns and highlighting the significance of our contributions. We are grateful for the reviewer’s time and effort in evaluating our work and remain open to address any further questions.

---

### Official Review · Reviewer_jkYR · 2024-11-03

**Soundness:** 3
**Presentation:** 4
**Contribution:** 3
**Rating:** 6
**Confidence:** 3

**Summary:**

The paper introduces Credit-Based Self-Organizing Maps (CB-SOM), a novel algorithm designed to integrate topographical organization into deep neural networks by aligning with top-down learning processes in DNNs. This new method modifies the traditional Kohonen’s Self-Organizing Map (SOM) to assign credit based on each unit's contribution to minimizing task loss. CB-SOM significantly improves object recognition performance compared to prior topographical models while maintaining alignment with the ventral visual cortex of macaques and humans. The model enhances representational alignment with neural activity in both early and high-level visual cortices, displaying substantial improvements over previous approaches.

**Strengths:**

1. The paper addresses the intriguing challenge of integrating topographical organization into deep neural networks, a problem with significant implications for both artificial intelligence and neuroscience.

2. The authors clearly articulate the motivation behind their work, highlighting the need to reconcile the performance of self-organized neural networks with the functional efficacy observed in biological systems.

3.  A comprehensive literature review provides context and demonstrates the paper's grounding in existing research.

4. The paper offers a thorough explanation of the proposed CB-SOM algorithm, including a detailed summary of the parameters for the experiments.

5. The authors effectively link their findings to supporting papers, relating the observed behaviors of their model to biological phenomena, thereby strengthening their claims.

6. The results showcase the superior quality of learned representations in CB-SOM and its enhanced alignment with neural responses in both human and non-human primate brains, demonstrating significant improvements over previous models.

**Weaknesses:**

1. The paper does not adequately assess / estimate the efficiency and computational time required for the CB-SOM learning algorithm, which could be a critical factor for its practical application.

2. The 'main contributions' statement in the introduction claims substantial improvements in object recognition performance vs prior topographical NNs, but does not comment on performance degradation compared to non-topographical models like ResNet. Including a comment on this trade-off would clarify the performance implications.

3. The trade-off in performance (as seen in Figure 2A) may affect the algorithm's appeal for studies outside neuroscience. Additional commentary from the authors on this would be beneficial.

4. The "Deep topographical neural network" section would benefit from a high-level summary of how the previous approaches in the literature impact overall performance, providing clearer context for readers.

5. It would be essential to provide a link to the code to ensure reproducibility

6. Figure 2A should include comparison with the other baselines shown in Figure 2B,for completeness

**Questions:**

Minor:

1. It would be interesting to discuss implications of this claim ‘resulting CB-SOM model displays substantial improvements in representational alignment with recordings from macaque visual cortex and imaging data from human visual cortex’

2. I am curious if you can provide some concrete examples for this claim ‘we believe our results are not limited to this task and could potentially be generalized to other tasks and contexts that may include multiple sensory modalities.’

3. A few typos e.g. ‘of of’ in line 510 and ‘th’ in line 205

---

> ### Author Response · Authors · 2024-11-21
> **Response to initial comments 1/2**
>
> We thank the reviewer for their thoughtful comments and recognition of our work’s strengths, particularly the novelty of CB-SOM in integrating topographical organization into deep neural networks and its implications for neuroscience and AI. We have carefully considered the reviewer’s questions and comments and provide detailed responses addressing each point:
>
> **Comparison of computational efficiency.**
> Thank you for your comment. We would like to first emphasize that as the architecture of the model used in our experiments is the same as the original model (ResNet18), the computational cost of our proposed CB-SOM model is exactly the same as the original ResNet18 model architecture at \emph{inference time}. The competitive algorithm is only used during training and does not apply at inference time. In that regard, the only difference in computational cost is during training as we introduce additional steps for selecting BMUs and performing the SOM-based weight updates. For transparency, we measured and compared the training step-times for each of the baselines and added the information in a new appendix Figure A5.
>
> **Highlighting the remaining gap in performance.**
> To address the reviewer’s concern, we updated the “main contributions” statement to better reflect this remaining challenge. We further updated the “discussions and limitations” section to appropriately emphasize the significance of the remaining gap (line 537).
>
> **Impact of the performance trade-off on the algorithm’s appeal outside neuroscience.**
> Thank you for the comment. We acknowledge that the lower performance of topographical models may make them less attractive as practical tools. We would like to emphasize that the motivation of our study is to replicate a missing characteristic of the cortex in existing deep neural networks, cortex-like topographical organization, in order to improve the existing models of visual cortex. This is an important characteristic missing from existing models that our project aims to address.  Topographical neural network models are thought to be potentially immensely useful for guiding therapeutic interventions such as stimulation or surgeries [1]. For example, planning resection surgeries in patients with epilepsy could benefit from predictions from topographical models that match the organization of the cortex as surgeons often plan their surgeries to avoid cortical patches that serve critical functions such as face recognition.
> We updated the introduction and discussion sections to better incorporate the motivation as well as potential benefits of our research (lines 50-58, 515-522).
>
> [1] Schrimpf, M., McGrath, P., Margalit, E., & DiCarlo, J. J. (2024). Do Topographic Deep ANN Models of the Primate Ventral Stream Predict the Perceptual Effects of Direct IT Cortical Interventions?. bioRxiv, 2024-01.
>
> **High-level summary of performance drop in previous approaches.**
> We appreciate the reviewer’s comment and agree that including this information would provide better context to our readers. We revised the “Deep topographical neural network” subsection and reported the range of relative drop in object recognition accuracy based on prior models of cortical topography (lines 153-154).
>
> **Providing code for reproducibility.**
> We included all the code required for replicating our results in the supplementary material. We will also publicly release the code upon publication.
>
> **Other baselines in Fig2A.**
> To address the reviewer’s concern, we added the non-topographical ResNet18 architecture to Fig 2a as an additional baseline. We did not include the TDANN model as the supervised-trained variation of this model does not elicit strong topographical organization and only the unsupervised trained variation replicates topography. However, we felt that it would be unfair to compare the learning speed between models that are trained, supervised and unsupervised.
>
> **Discussing the implications of improved brain alignment in CB-SOM.**
> We appreciate the reviewer’s comment. Our results show that in particular when compared to data from the human visual cortex, our topographical CB-SOM model is superior to the non-topographical ResNet18 model. This result suggests that inducing topographical organization in deep neural networks through overrepresentation of units with maximal effect on task performance (similar to CB-SOM) may lead to more aligned representations with the brain. We revised the Discussions section and expanded on the implications of these results (lines 496-500).
>
> (Continues in the next comment)

---

> > ### Author Response · Authors · 2024-11-21
> > **Response to initial comments 2/2**
> >
> > (Continued from previous comment)
> >
> > **Generalization to other tasks and modalities.**
> > We primarily base this prediction on several recent work that demonstrated the applicability of topography-inducing approaches first developed for vision models to language models (e.g [1,2]). In these other works which considered the transformer architecture, the attention and MLP units were similarly arranged on a 2-dimensional map and topography was induced such that nearby units on these maps had more similar response profiles. We added this information to the “discussion and conclusion” section to further support our claim (lines 483-487).
> >
> > [1] Rathi, N., Mehrer, J., AlKhamissi, B., Binhuraib, T., Blauch, N. M., & Schrimpf, M. (2024). TopoLM: brain-like spatio-functional organization in a topographic language model. arXiv preprint arXiv:2410.11516.
> >
> > [2] Binhuraib, T. O. A., Tuckute, G., & Blauch, N. (2024, March). Topoformer: brain-like topographic organization in Transformer language models through spatial querying and reweighting. In ICLR 2024 Workshop on Representational Alignment.
> >
> >
> > **Typos.**
> > We thank the reviewer for bringing these issues to our attention. We corrected the typos in our revised manuscript.

---

> ### Author Response · Authors · 2024-11-27
> **Further discussions and clarifications**
>
> Dear Reviewer,
>
> Thank you again for taking the time to review our work. We’ve provided detailed responses to your comments and would greatly appreciate any further feedback or clarifications from you regarding our rebuttal. Your input is invaluable in helping us improve our manuscript, and we are eager to engage in a constructive discussion. Please let us know if there are any additional points we can address.
>
> Thank you for your time and consideration!
>
> Authors

---

> > ### Comment · Reviewer_jkYR · 2024-11-28
> >
> > I thank the reviewers for addressing my questions and concerns thoroughly and for revising the paper accordingly. Based on these improvements, I will raise my score by one point, now assigning it a 7.

---

> > > ### Author Response · Authors · 2024-11-28
> > > **thank you!**
> > >
> > > Thank you for taking the time to provide thoughtful feedback on our work and for raising your score. We sincerely appreciate your engagement and consideration which has significantly improved the quality of our work.

---

### Author Response · Authors · 2024-11-21
**General response to all reviewers**

We sincerely thank the reviewers for their insightful and positive feedback. We are encouraged by the reviewers’ acknowledgement of the novelty of our work (reviewers **jkYR**,**WLW9**, and **aFS5**), clarity of writing (reviewers **jkYR**,**nVAR**, and **aFS5**), rigorousness and thoroughness of our work (reviewers **jkYR**, **WLW9**, **nVAR**, and **aFS5**), and impactfulness of our results (reviewers **jkYR** and **WLW9**), while identifying unclarities and limitations. Below, we point out several common questions and the major updates that we have done as a result of the reviews. We respond to all inquiries in individual responses.

- **Clarifying the motivation and benefits of topographical models** (reviewers **jkYR**, **aFS5**). We revised the introduction and discussion sections to more clearly explain the motivation behind building topographical models and discussing the potential benefits of these models in both neuroscience and AI.

- **Project code for reproducibility** (reviewer **jkYR**). We provided the project code in the supplementary material.

- **Comparison of computational efficiency** (reviewer **jkYR**). We compared the computational cost of training each model variation in a new Fig. A5 in the appendix.

- **Alternative architecture in results and appendix** (reviewer **WLW9**). We additionally trained the CORnet-S model using our proposed CB-SOM method and showed that this model also demonstrates topographically organized features in both its early and deep layers. We showed these results in the new Fig. A6 in the appendix.

- **Updated Figures to add more baselines and clarification** (reviewers **jkYR,WLW9,aFS5**). We updated Figs. 1,2, and 4 and added more baselines requested by the reviewers and clarified what units on the 2D simulated cortical sheet correspond to.

- **Relation to cortical lateral connections** (reviewer **WLW9**). We added a new section in the appendix that discussed in-depth the relation between the SOM update rule as used in our work as well as the notion of local lateral connections in the brain.

- **Revise the Discussion and Conclusion section to add limitations** (reviewers **jkYR,aFS5**). We added a new Limitations subsection in the “Discussions” section and included an extended discussion of the limitations of our work including “the Lack of formal convergence guarantees” and “Remaining performance gap between topographical and non-topographical models”.

---

### Meta-Review · Area_Chair_iR3i · 2024-12-20

**Metareview:**

The paper presents an interesting and novel approach for integrating topographical organization into deep neural networks and narrowing the performance gap that has historically hindered self-organized topographical models. The authors provide a credit-based self-organizing map mechanism that aligns well with top-down learning signals, producing results that closely parallel characteristics of the primate visual ventral stream without dramatically sacrificing performance. Given the overall positive reception from multiple reviewers and the addressed concerns, the paper should be accepted.

**Additional Comments On Reviewer Discussion:**

Reviewers generally found the manuscript to be well-written, solid in methodology, and thorough in its empirical comparisons, noting the important contribution of bridging neuroscience-inspired topographical constraints with practical deep learning architectures. While some reviewers requested further clarification on certain details and broader motivations, the authors responded constructively, strengthening the manuscript and adding relevant experiments, including training an additional architecture and clarifying the functional relevance of their approach.

---

### Decision · Program_Chairs · 2025-01-22

Accept (Poster)